# Participation in Intergenerational Food and Agriculture Education Programs Effectively Promotes Place Attachment

**DOI:** 10.3390/ijerph20054616

**Published:** 2023-03-05

**Authors:** Po-Ching Wang, Jing-Wun Huang, De-Chih Lee

**Affiliations:** 1Department of Landscape Architecture, National Chiayi University, 300 Syuefu Rd., Chiayi City 600355, Taiwan; 2Zhuqi Elementary School, Chiayi County 621301, Taiwan; 3Department of Information Management, Da-Yeh University, Changhua City 515006, Taiwan

**Keywords:** intergenerational education, food and agricultural education, place attachment, place identity, place dependence

## Abstract

This study adopted an empirical approach to examine the effectiveness of integrating intergenerational education with food and agricultural education to increase students’ affection for their learning environment. The intergenerational food and agricultural education program in this study consisted of various courses promoting educational dialogue between students and their parents and grandparents at home. The bidirectional learning process allowed the three generations to better understand each other’s dietary and life experiences and pass on the relevant knowledge and culture. The 51 participants in this quantitative study were rural elementary schoolchildren who were divided into an experimental group and a control group. Place attachment was evaluated through the two sub-dimensions of place identity and place dependence. The results revealed that food and agricultural education implemented as intergenerational education strengthens learners’ affective attachment to their school environment.

## 1. Introduction

Food and agricultural education (FAE) is a neoteric and important issue. FAE allows learners to gain practical experience in food production and consumption-related processes [1]. By participating in FAE programs, learners experience the ecological, economic, and cultural factors of food systems and develop their health promotion literacy.

The goal of FAE is to establish the relationships between humans and foods, lands, and cultures so that learners understand the origins of the food they consume as well as the value of agricultural activities. Due to the steady rise of industrialization and urbanization from the 18th and 19th centuries onward, many populations around the world consist of urban dwellers [2]. As a result of these changes in society, younger generations often do not understand how the foods they consume are produced [3]. Their ignorance of food-related issues has led to problems including imbalanced nutrition and food waste [4]. Furthermore, because food education has been gradually commercialized by private enterprises, the public only has access to limited and specific information, making it difficult for them to improve their knowledge of healthy food culture [5].

Despite the remarkable benefits of FAE, there are barriers to the practical implementation of FAE programs. For example, many schools lack the teaching specialty, funding, and learning spaces; have long curriculum preparation times; encounter difficulties in crop management; and have no target crops. However, in practice, intergenerational assistance and cooperation may mitigate teachers’ teaching stress and help learners spread FAE to their families. In turn, learners share their experiences at home in the classroom. As a result, FAE becomes a part of the learners’ daily lives, nurtures intergenerational affection, and strengthens learners’ care and concern toward their learning spaces. By integrating intergenerational education with FAE, this study conceived the concept of intergenerational FAE (IFAE).

### 1.1. The Significance and Educational Goals of FAE

FAE consists of food education, nutrition education, and farm experiences [6]. As a result of drastic changes to the landscape and structure of rural villages caused by urbanization, the tradition of preserving rural dwellers’ identification with their farmland settings and farming activities has gradually faded. To address this issue, many countries have begun to promote FAE in elementary schools. Based on the outcomes, learners have gained diverse and interesting learning experiences that have allowed them to change their perspectives pragmatically and positively [7]. Exposing children to FAE at a young age and integrating environmental and life sciences courses enables the children to practice wholesome dietary habits and pass on food cultures to future generations.

Throughout the course of human history, farming activities have been regarded as the activities most directly associated with the natural environment and our livelihoods. However, industrialization and urban population growth have separated agriculture from living environments. Land is viewed as a production tool, and farmers mostly generate produce to earn profits. Their pursuit of quick, mass, and intensive methods to produce crops has ultimately deteriorated the environment and reduced biodiversity [8,9]. In light of these concerns, urban dwellers and children who grew up in a rural backdrop should reestablish their knowledge of agricultural activities in order to understand the role that humans play in natural environments. FAE is an excellent tool and medium to achieve this.

FAE compensates for the sense of detachment between theoretical education and environmental issues in reality. As students engage in laborious and hands-on tasks, they apply their acquired knowledge, skills, and attitudes in their lives. Thus, FAE courses rebuild mankind’s relationship with the land and food.

### 1.2. The Health Benefits of FAE

FAE activities are salubrious. A person’s dietary habits and choices are strongly influenced by their childhood; participating in FAE programs at school provides more opportunities for children to consume fruit and vegetables and promotes positive attitudes and behaviors related to maintaining a balanced diet, thus reducing their likelihood of developing chronic diseases during adulthood. The agriculture and horticulture spaces also provide experiential learning opportunities with regard to the basic prevention and treatment of diseases and have therefore become a component of medical insurance-related experiences. FAE thus helps learners construct healthy dietary habits. By growing fresh produce themselves, learners develop their nutrition and horticultural knowledge, thus changing their lifestyles for the better [10]. Lifestyle habits often change with work patterns. In the post-industrialization era, most people sit for long hours at work and home. The deleterious combination of decreased activity levels and greater intake of irritating, sugary, high-fat, and high-calorie foods for stress relief not only causes a person to become overweight, but is also associated with chronic diseases including type 2 diabetes, hypertension, cardiac disease, and obesity [11]. Nutritional interventions play a crucial role in addressing these health problems, and adopting healthy dietary patterns is a primary means of illness prevention [12,13]. The dietary behaviors of most adults are influenced by advertisements, the accessibility of foods, and their dietary habits and concepts conceived during childhood [1]. Therefore, schools should be obliged to provide nutrition education to promote students’ healthy eating habits. Promoting FAE and nutrition education at school improves students’ length of exposure to and experiences with agricultural crops and enhances their preference for fruits and vegetables [14]. Nutrition education that includes creating community vegetable gardens also improves toddlers’ willingness to try fruits and vegetables, increases adolescents’ intake of fruits and vegetables, and strengthens adults’ dietary status and health [15]. Moreover, participating in suitable farming activities is therapeutic for the mind and body, as practical experiences in crop and plant growing significantly improve one’s activity levels, positive emotions, and physical and mental health [16,17,18].

### 1.3. The Significance and Educational Goals of Intergenerational Education

Intergenerational education is an educational form in which learners acquire learning opportunities through interactions between different (two or more) generations. Berenbaum and Zweibach [19] pointed out that intergenerational education allows different generations to connect with one another by exchanging values, resources, traditional cultures, and cross-generational wisdom. In addition to receiving education in schools, children engage in interactions with their family members and elders that function as a means of imparting knowledge, techniques, and emotional support. However, in social contexts marked by expanding population structures, middle-aged individuals are often burnt out from providing care to their elders and parenting their children. Family dysfunction then occurs because family members are unable to interact and communicate in a jovial manner [20]. Intergenerational education is among the many proposals and methods that have been proposed to overcome this social issue. In this type of education, instructors act as facilitators in a series of learning activities that foster intergenerational relationships. As learners share values, resources, and cultures, they gain opportunities to engage in mutual care and cross-generational exchanges [21]. In addition to promoting intergenerational relationships between elderly and young generations, intergenerational education encourages elderly learners’ self-identification as they support their younger counterparts’ learning process, while young learners improve their competence in overcoming adversities through the shared experiences and encouragement received [22]. The goal of intergenerational education is to accumulate and impart knowledge and to proactively promote a rapport between different cohorts, thus creating harmonious social and psychological environments that serve as the basis for the social safety net. In practice, intergenerational education alleviates schoolteachers’ stress and improves their concern toward their teaching environments.

### 1.4. Place Attachment

Place attachment is a concept in environmental psychology that was first coined in the late 20th century [23,24]. Place attachment stems from an individual’s behavioral and affective responses to events they have experienced in a specific environment. Subsequently, the individual forms place identity and place dependence and establishes the meaning of the place to them. Place dependence is created when an individual has specific functional needs from a particular place, while place identity is created when they develop a specific affection for a place [25,26,27]. Place here refers to settings where meaning is conveyed through individual, group, or cultural processes [28]. Participating in FAE activities enhances the physical performance and learning motivation of children and adolescents. Students with excellent learning attitudes proactively learn agriculture-related knowledge and skills and thus enhance their place attachment to their school. When students apply themselves to FAE programs, they are more attentive, invested, and happy during the experiential courses. As they understand and experience more about their educational environment, they develop complex affective bonds as well as a sense of attachment with time [29]. Since environment attachment reflects one’s specific affection for a place, one of the best approaches to community environmental management may be promoting environment attachment [30]. This approach can improve neighborly relations, facilitate the implementation of local affairs [31,32], and enhance people’s willingness to perform environmentally friendly and environmental conservation behaviors, including those promoting ecological resource sustainability [11].

### 1.5. The Potential Influence of IFAE on Environment Attachment

During an intergenerational education program, students bring the FAE curriculum home and engage in dialogue with their elders, thus promoting conversations across different generations and boosting the elders’ confidence. The students strengthen their familial relationships and receive wisdom and culture passed on by their parents and grandparents through bidirectional learning across three generations. Rewarding outcomes are gained as long as periodic observations and experiences occur in specific environments. Interacting with others during the activities also elicits numerous affective responses, thereby improving the affective cohesion between different generations as well as the participants’ impression of and affective bonding with their surroundings [33]. Relevant studies have shown that regular or seasonal intergenerational activities (including harvesting, horticulture, and hunting) and eventful group activities (such as rebuilding after disasters and international competitions) provide a means to transmit knowledge, techniques, and life experiences between generations through important objects or spatial memory [34,35,36]. The common objectives of the intergenerational activities also create an affective rapport between the participants that strengthens the bond between different generations, enabling the participants to form lasting impressions of their environments and gain a sense of place attachment and place identity [37]. Intergenerational activities not only strengthen social groups and form social support systems but also make up for the differences between groups and social tiers and contribute positively to social development.

### 1.6. The Objective of Study

The objective of this study is to examine whether IFAE may strengthen students’ affection for their learning environment more than FAE. It is expected that the experiential IFAE programs implemented at the school have a profound influence on students, helping them care more about nature and their surroundings and clearly understand their responsibilities and the food cultures around them. Such programs shape students’ environmental awareness, competence in developing solutions for environmental and health problems, and interest in local community affairs.

## 2. Materials and Methods

### 2.1. Study Framework

The hypothesis (H1) of this study posits that:

**Hypothesis** **1** **(H1).** *Applying intergenerational education as an intervention enhances FAE participants’ affection for their learning environment*.

Based on the literature review, we developed the study framework as shown in Figure 1.

### 2.2. Study Area and Participants

The target participants of this program were elementary schoolchildren in the rural regions of Zhongpu Township, Chiayi County, Taiwan (see Figure 2). Since the school only has six classes, almost all students participated in this study. The students were divided into an experimental group (received intergenerational education) and a control group (did not receive intergenerational education). In line with the number of samples recommended by the central limit theorem for normal distribution, around 30 participants were expected in each group. Zhongpu Township is an important agricultural township in Taiwan. The schoolchildren there come from multigenerational farmer families. The advantage of implementing FAE or IFAE at the township’s elementary schools is that the land cultivated and the farming methods applied by the school are in tune with the actual daily lives of the children.

### 2.3. Study Instruments and Course Design

This study was carried out in line with the school’s curriculum as well as the Rice Education Program designed by the Council of Agriculture. To examine the influence of IFAE program participation on environment attachment, the teaching plan was designed to have two parts: general FAE and intergenerational FAE. The program theme was “Enjoy Your Food Every Time”, and it consisted of three stages: Food for Meaning, Food for Planting, and Food for Skills Sharing (Figure 3). Intergenerational FAE was incorporated into each stage with the purpose of encouraging the students to engage in dialogue with their elders and complete worksheets. At the end of the course, the participants anonymously completed our self-developed Environment Attachment Questionnaire.

The farming courses and experiential activities included in this program depend on the climate, environment, and season, and the educational contents are revised and designed accordingly. The course hours of the FAE and IFAE programs were both flexible. Students received their own plots of farmland from the school’s Tasty Garden farming project and engaged in hands-on experience farming activities such as plowing, rice growing, weeding, and pest control. By touching real soil, the students played the role of mini farmers and renewed their understanding of the land where they were born and grew up.

In the first stage, which was titled Food for Meaning, the students grew crops and learned about the edible parts of various vegetables, as well as the local customs and dietary habits of different countries. The experimental group (intergenerational education program participants) had an additional worksheet requiring them to discuss an unforgettable family dish the elder family members had enjoyed in their childhood, and then, with help from their elder family members, design a recipe with a story.

In the second stage, which was titled Food for Planting, the students experienced the process of caring for their crops. They learned about crop rotation for growing vegetables, the suitable seasons and methods for growing different fruits and vegetables, existing agricultural environments, the lower levels of pesticide residue in in-season fruits and vegetables, and the eco-friendly practice of adopting a low-carbon diet. The experimental group had an additional worksheet that required them to discuss with their elders the differences in the agricultural environments of their generations so that they could understand the soil-friendly practice of natural farming.

In the third stage, which was titled Food for Skills Sharing, the students prepared their own dishes and learned about the relationships between food and nutrients, the nutrients they need, as well as the influence of maintaining a balanced diet on their health. The experimental group had an additional worksheet, which required them to learn from their elders how to prepare a dish and then share their experiences with their classmates.

### 2.4. Questionnaire Scale

This research adapted Jorgenson and Stedman’s [38] Sense of Place scale into the Place Attachment Questionnaire that was administered to the students. The items were measured on a seven-point Likert scale ranging from strongly disagree (1 point) to strongly agree (7 points). The questionnaire covered six items on place identity, eight items on place dependence, and the participants’ socioeconomic backgrounds.

### 2.5. Data Analysis Approaches

Invalid questionnaire responses, such as incomplete or duplicate responses, were removed to ensure data accuracy. The data were recorded and analyzed using AMOS 24.0 and SPSS 26.0 software. The statistical methods included descriptive statistics, reliability and validity analysis, normality test, confirmatory factor analysis, and independent sample *t*-tests.

## 3. Results

A total of 62 questionnaires were distributed in this study, and 51 valid samples were collected after deducting 5 incomplete questionnaires and 6 that were invalid because respondents had selected multiple responses on the Likert scale to individual questionnaire items. Questionnaire items with a moderate corrected item-total correlation (<0.03) were removed. Five items pertaining to place identity, and four items pertaining to place dependence were retained. The reliability analysis results showed that the Cronbach’s α of place identity, place dependence, and place attachment was 0.715, 0.783, and 0.857, respectively. As all these values were greater than 0.7, the internal consistency of the items was good, and the scale had excellent reliability. Next, AMOS was used for confirmatory factor analysis. Since only two items had a standardized regression coefficient greater than 0.7, we decided to apply a less stringent standard and retained all items with a standardized regression coefficient greater than 0.5.

Regarding the convergent validity of the scale, the composite reliability (CR) of place identity and place dependence was 0.734 and 0.756, respectively, while the average variance extracted (AVE) of place identity and place dependence was 0.357 and 0.448, respectively. All the CRs exceeded 0.7 and were acceptable [39], while the AVEs were rather low. Regarding the discriminant validity of the scale, the correlation coefficients of the dimensions at a 95% confidence interval were calculated using the bootstrap method with 1000 bootstrap samples. The results showed that the correlation coefficients of place identity and place dependence at a 95% confidence interval were 0.077 and 0.729, respectively. Since 1 was not included in the confidence interval, the scale had good discriminant validity [40].

In this study, AMOS 24.0 software was adopted to analyze whether the collected data fit the normal distribution. The value of Mardia’s coefficients of multivariate skewness and kurtosis was 52.050 (<99, based on 9 × 11) [39], indicating that the data satisfied the multivariate normal distribution.

The model fit was evaluated using the recommendations proposed by Jackson, Gillaspy, and Purc-Stephenson [41] in their 2009 review of 194 Social Science Citation Index (SSCI) studies. The results of five model fit indices were as follows: the overall fit index (χ^2^/df) was 1.769 (the ideal value should not exceed 3); the goodness of fit index (GFI) was 0.861; the adjusted goodness of fit (AGFI) index was 0.727; the root mean square residual (RMR) was 0.102; and the standardized root mean square residual (SRMR) was 0.078. Generally speaking, the model fit was good, as shown in Table 1.

### 3.1. Description of the Sample

Based on the effective sample size of 51, there were 25 students (9 boys, 16 girls) in the IFAE program (experimental group) and 26 students (13 boys, 13 girls) in the general FAE program (control group). There were 29 (57%) female and 22 (43%) male students. Second graders accounted for the majority of the sample at 33%, followed by sixth graders at 18%. The majority (39%) of the participants’ fathers worked in agriculture, forestry, fishing, and animal husbandry, while their mothers were mostly (27%) not employed outside their homes. The highest level of education completed by most of the students’ parents was vocational/senior high school education (59% for fathers and 49% for mothers), as shown in Table 2.

### 3.2. Place Identity, Place Dependence, and Place Attachment Levels of the Experimental and Control Groups

This study employed independent sample *t*-tests to compare whether the experimental and control groups differed with regard to their mean levels of place attachment and its sub-dimensions of place identity and place dependence. The experimental group (IFAE) consisted of 25 students, while the control group (FAE) consisted of 26.

The results showed that the experimental group’s mean levels of place attachment, place identity, and place dependence were all greater than those of the control group. The mean level of place attachment of the experimental and control groups was 6.440 and 5.915, respectively; the mean level of place identity of the experimental and control groups was 6.600 and 5.384, respectively; and the mean level of place dependence of the experimental and control groups was 6.312 and 5.977, respectively. The homogeneity of variance test results revealed that, with the exception of place identity, the variables (place dependence and place attachment) met the assumption of equal variance.

As shown in Table 3, the independent sample *t*-test results of the experimental and control groups showed that the mean differences in place attachment and place identity were significant. The mean level of place attachment differed significantly between the experimental and control groups (*F* = 0.725, *t* = 2.410), which meant that compared to the general FAE program, the students significantly improved their level of place attachment after participating in the intergenerational FAE program. The mean level of place identity also differed significantly between the experimental and control groups (*F* = 5.100, *t* = 3.268), which meant that compared to the general FAE program, the students significantly improved their level of place identity after participating in the IFAE program. However, there were no significant differences (*F* = 0.052, *t* = 1.446) in the mean levels of place dependence of the control and experimental groups.

## 4. Discussion

Based on the data of this research, students who participated in general FAE (control group) and students who participated in IFAE (experimental group) both improved their level of place attachment to the school settings. Related studies pointed out that if an individual generates special emotions or functional needs in relation to a space, the individual may develop a sense of place within the space [25]. During the teaching process of this course, students engaging in their favorite planting and cooking experience courses generated a positive psychological state in the educational environment. FAE helped students explore and feel the campus environment with which they were in contact. Over time, and as they participated in the courses, students developed multiple emotions in relation to the campus environment. As argued by Chapin and Knapp [30], environmental attachments reflect the special meanings, values, and individual feelings that students associate with places. These connections to places transformed over time into emotions as they were felt, imagined, identified, and understood. Due to the significant benefits of FAE, many countries are actively promoting related educational programs. Government agencies are increasingly encouraging schools and communities to develop FAE to help improve schoolchildren’s eating habits, food choices, cooking skills, nutritional knowledge, and social skills.

Importantly, this study demonstrated that IFAE is more effective than general FAE, although both IFAE and FAE can promote students’ attachment to campus settings. Related theories show that intergenerational programs may enhance the exchange of culture and values across generations and strengthen emotions [19]. Engaging in intergenerational activities creates a common goal, which stimulates emotional resonance and strengthens mutual emotional bonds. In addition, the participants form deeper impressions of their environmental space and thereby obtain place identity and place dependence [37]. In this study, the theory of intergenerational education was practiced. Students brought the content of the IFEA course home to their families and engaged in dialogue with their elders. The intervention of IFEA therefore helped the students and their elders share life wisdom and culture and enhance emotional communication. The participation of their elders in the teaching encouraged the students to absorb traditional experience and skills. In addition, the intervention made the students more interested in the courses and the campus setting, which created more cohesion between the students and their environment.

According to Vaske and Kobrin [11], environmental attachment refers to the cohesive relationship between the individual and the environment, which includes two sub-dimensions: place identity (the emotional attachment) and place dependence (the functional attachment). This study expected that through IFEA, schoolchildren would better identify with their culture and become closer to the land, thereby strengthening the children’s sense of attachment to their school and community. These theories were partially verified in this study. The difference in the level of place dependence was not significant, probably because the test environment was an agricultural setting (the school grounds were surrounded by plots of farmland). The students also had a certain degree of familiarity with the planting environment. Therefore, their sense of dependence on facilities and settings in school planting areas was less likely to be influenced by intergenerational education.

The reliability and validity of the results of this study would have been improved had the sample size been larger. It was originally expected that the control and experimental groups would each have 30 subjects. Although there were only 25 (control group) and 26 (experimental group) subjects in the end, the number was still close to the value suggested by the central limit theorem. The test of skewness and kurtosis also confirmed that the data conform to the multivariate normal distribution. Hence, the research results are still convincing.

## 5. Conclusions

This study demonstrated that implementing intergenerational education practices in an FAE program was more effective in evoking the participants’ place attachment to their learning environment compared to a general FAE program. Therefore, we suggest that schools integrate intergenerational education practices into their curricula related to school environment education. The systematic planning and implementation of such curricula enhances students’ unique affection for and identification with their school environment, thereby nurturing their sense of attachment. It is also suggested that such programs should be promoted in high schools to help young people [42].

Because of time, human resources, and material constraints, this study was only able to implement the IFAE and general FAE programs in a single elementary school. The sample size was relatively small due to the limited student population of the school. Therefore, we recommend that future studies enlarge the study scope and include more students. In addition, the IFAE program was implemented through worksheets which the students discussed with their elders at home. It is possible that the elementary school students had not yet developed mature expression skills and were thus unable to express their opinions clearly. Thus, future studies can organize direct parent–teacher discussions to acquire more of the students’ background information and improve the value of the study results. It is also recommended that schoolchildren are given more assistance in completing the questionnaires to reduce the number of invalid questionnaires obtained. In addition, IFAE research could be conducted in urban areas [43] or among students who have less agricultural knowledge and experience in order to further explore place dependence among these students.

## Figures and Tables

**Figure 1 ijerph-20-04616-f001:**
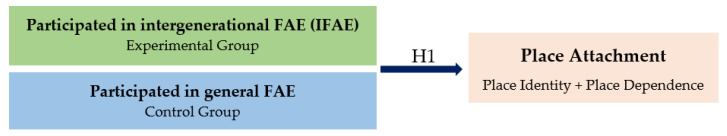
The study framework.

**Figure 2 ijerph-20-04616-f002:**
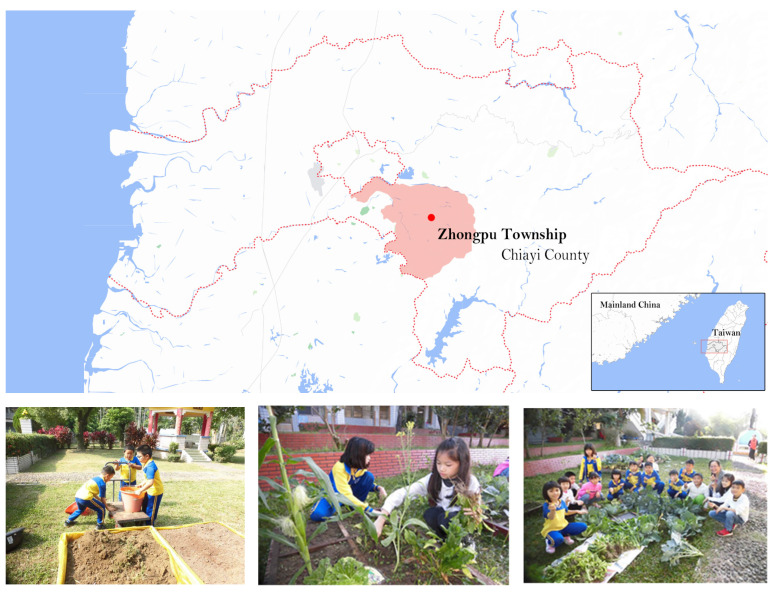
Location of the study area (**upper**) and students participating in the study (**lower**).

**Figure 3 ijerph-20-04616-f003:**
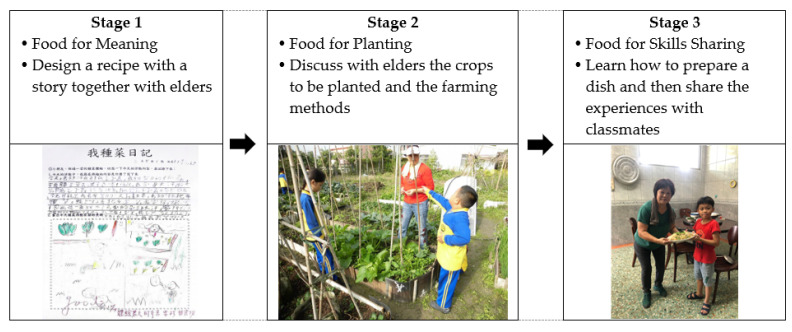
The three stages of integrating intergenerational education into the FAE program.

**Table 1 ijerph-20-04616-t001:** Summary of the results of the reliability and validity analysis of the Environment Attachment Questionnaire.

Dimension	Item	Corrected Item Total Correlation	Standardized Regression Coefficient	Reliability and Validity
Place Identity	I perceive myself as a member of the school.	0.545	0.687	Cronbach’s α = 0.715 CR = 0.734 AVE = 0.357
	To me, school is a particularly important place.	0.476	0.555	
	I strongly agree with the curriculum arranged by the school.	0.414	0.564	
	To me, school is particularly meaningful.	0.587	0.656	
	To me, school is extremely important.	0.634	0.612	
Place Dependence	I gain greater satisfaction at school than at other places.	0.620	0.514	Cronbach’s α = 0.783 CR = 0.756 AVE = 0.448
	To me, learning at school is more important than other things.	0.689	0.793	
	I enjoy sharing the events at school with others.	0.512	0.501	
	To me, reading at school is more important than reading in other places.	0.555	0.803	
	Overall Cronbach’s α = 0.857 Model Fit Indices: $\chi^{2}/DF=1.769$; GFI = 0.861; AGFI = 0.727; RMR = 0.102; SRMR = 0.078			

**Table 2 ijerph-20-04616-t002:** Descriptive statistics of the sample (N = 51).

Variable	Category	Frequency (Persons)	Percentage (%)	Variable	Category	Frequency (Persons)	Percentage (%)
Sex	Male	22	43%	Father’s occupation	Agriculture, forestry, fishing, and animal husbandry	20	39%
	Female	29	57%		Industry	13	25%
Grade	First grade	8	16%		Commerce	1	2%
	Second grade	17	33%		Military, police, civil servant, teacher	1	2%
	Third grade	6	12%		Service industry	6	12%
	Fourth grade	5	10%		Professional technician	7	14%
	Fifth grade	6	12%		Other	3	6%
	Sixth grade	9	18%	Mother’s occupation	Agriculture, forestry, fishing, and animal husbandry	9	18%
Father’s level of education	Elementary school or below	8	16%		Industry	5	10%
	Senior or vocational high school	30	59%		Commerce	10	20%
	Junior college or university	13	25%		Military, police, civil servant, teacher	3	6%
Mother’s level of education	Elementary school or below	5	10%		Service industry	9	18%
	Senior or vocational high school	25	49%		Professional technician	1	2%
	Junior college or university	21	41%		Other	14	27%

**Table 3 ijerph-20-04616-t003:** Summary of the *t*-test results of the experimental and control groups’ place attachment, place identity, and place dependence.

Dimension	Group	Sample Size (n)	Mean (*M*)	Standard Deviation (*SD*)	Levene’s Test (*F*-Statistic)	*t*-Statistic
Place Attachment	Control	26	5.915	0.832	0.725	2.410 *
	Experimental	25	6.440	0.718		
*Place* *Identity*	Control	26	5.384	0.985	5.100 *	3.268 **
	Experimental	25	6.600	0.657		
*Place* *Dependence*	Control	26	5.977	0.813	0.052	1.446
	Experimental	25	6.312	0.843		

Note: * indicates *p* < 0.05; ** indicates *p* < 0.01.

## Data Availability

The data are available upon request via email or phone to the corresponding author.

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
