# Peer review of "Participation in Intergenerational Food and Agriculture Education Programs Effectively Promotes Place Attachment"

_ijerph, 2023, doi:10.3390/ijerph20054616_

Round 1

Reviewer 1 Report

This is a very interesting and important article. The authors make the case that for most people, we are far removed from the growing of our food. Farming in the US is very much geared toward profit with little appreciation for staying close to the land and treating the land and our food sources with respect and care. This article is compelling because through growing and preparing food it fosters appreciation for family (especially attachment to the elders), culture, and is an important way to honor the ancestors.

The literature review is strong and makes the case for the importance of this work.

I note that the study design and instruments appear to be very good. I am not a statistician and so I hope another reviewer can be helpful here. I must note that having only 6 boys in the experimental group is a bit of a concern but nothing can be done about that. Figure 3 is well-done and makes the work clear to the reader.

The findings that the experimental group showed deeper levels of place attachment, place identity, and place dependence are important. These things, of course, enrich people's lives. 

I'd like to see the authors add more to the Conclusion/Recommendations for future studies (section 5.2). Please consider whether this work is meaningful beyond the level of elementary school students. What about  high school level? I'd also like to see the authors give attention to developing this kind of program in other countries. This sample was already steeped in farming and this work could easily be replicated in farming areas around the world. And of course, what about encouraging schools in cities to be able to adapt this model for students who do not have farming knowledge and experience? If nothing else this kind of program could give students, parents, and teachers alike a deeper appreciation for nature.

A few small suggestions:

Line # 27 change "neoteric but important issue." to "neoteric and important issue."

Line # 220 "where they were born and grew up in" is better in removing "in" so the end of the sentence reads "where they were born and grew up."

Line #s 288-289. It a bit controversial to say that housewives are "unemployed." I think this sentence could be rewritten to state that most women were not employed outside of their homes.

Overall, a very good effort!

Reviewer 2 Report

ijerph-2125468 Review

Summary: This study explored the effect of intergenerational learning on place attachment to the school using a lesson on food and agriculture. The authors found that place identity and place attachment were stronger in students that participated in intergenerational learning than in students who did not. While the results of the study should be interpreted with caution due to the small sample size, it acts as a useful starting place for others who may be interested in the topic.

General Comments:

You provide an excellent background into FAE and its goals and benefits, place attachment, and intergenerational learning. Everything in these sections is both explained well and well cited. However, I would classify sections 2.1-2.5 as background, not materials and methods. I suggest that you remove the “materials and methods” headings and change the numbers of 2.1-2.5 to 1.1-1.5 as subheadings in the introduction.

When looking at the hypothesis, I was a little confused as to why you were looking at how intergenerational learning influenced place attachment to the school. I can see how it might strengthen attachment to homes or community, but it could be useful to have an extra sentence or two regarding why you think it will influence connections to the school.

The conclusion section could be expanded upon more. Please add more explanation as to what your findings mean and why they may have occurred. Consider comparing your findings to other, related studies to help explain the meaning of the findings.

You mentioned that one limitation is the sample size, which I agree with. With less than 30 students in each group, it is not very generalizable. Ways to strengthen this study might be to replicate it at other schools, as you mentioned. You might also consider supplementing your results with qualitative data. Interview some of the students and their guardians from both groups to get more information on how they felt the activities helped strengthen their connections to place. While I think that the study as written is fine, adding more data, whether qualitative or quantitative, would make it excellent. If you do choose to keep it as is, I would strongly suggest adding a justification for why your sample size is adequate for your purposes. 

Around 65% of your citations are from over 10 years ago. This is not necessarily a bad thing, but something to be aware of. Some older articles may be important to keep if they are seminal works, but it may be useful to see if any can be replaced with slightly more modern ones. I don’t think it’s a huge deal if these are the best options, but still something to keep in mind.

Specific Comments:

Figure 1. This figure states that the experimental group participated in FAE and the control group did not. But lines 202-203 say that both groups participated in FAE, but the experimental group was intergenerational, and the control group was general. Please correct whichever one was written in error.

Line 251. The sentence mentioned that invalid data were removed. What was considered invalid?

Line 288. It says that most mothers were unemployed, but the number listed is 27%. Wouldn’t most mothers be employed if only 27% were not?
